# Astragaloside IV Regulates cGAS-STING Signaling Pathway to Alleviate Immunosuppression Caused by PRRSV Infection

**DOI:** 10.3390/v15071586

**Published:** 2023-07-20

**Authors:** Ke Song, Jia-Ying Yu, Jiang Li, Miao Li, Lu-Yuan Peng, Peng-Fei Yi

**Affiliations:** College of Veterinary Medicine, Jilin University, Changchun 130062, China; songke20@mails.jlu.edu.cn (K.S.); yujiaying21@mails.jlu.edu.cn (J.-Y.Y.);

**Keywords:** AS-IV, PRRSV, cGAS-STING, immunosuppression, antiviral activity

## Abstract

Porcine reproductive and respiratory syndrome virus (PRRSV) poses a global threat to pig health and results in significant economic losses. Impaired innate and adaptive immune responses are evident during PRRSV infection. Cyclic GMP-AMP synthase (cGAS), a classical pattern recognition receptor recognizing mainly intracytoplasmic DNA, induces type I IFN responses through the cGAS-STING signaling pathway. It has also been demonstrated that cGAS-STING is involved in PRRSV infection. This study utilized the qRT-PCR, ELISA, and WB methods to examine the effects of Astragaloside IV (AS-IV) on the regulation of innate immune function and cGAS-STING signaling pathway in porcine alveolar macrophages. The results showed that AS-IV attenuated the decreased innate immune function caused by PRRSV infection, restored the inhibited cGAS-STING signaling pathway, and increased the expression of interferon, ultimately exerting antiviral effects. Moreover, these results suggest that AS-IV may be a promising candidate for a new anti-PRRSV antiviral, and its mechanism of action may provide insights for developing novel antiviral agents.

## 1. Introduction

Porcine reproductive and respiratory syndrome is a highly contagious infectious disease caused by the porcine reproductive and respiratory syndrome virus, characterized by reproductive disorders in pregnant sows and other respiratory diseases in pigs of all ages. It is an important immunosuppressive disease of pigs. PRRS has been first reported in the United States since 1987, and soon spread to countries around the world, causing an epidemic worldwide, plaguing the continuous and healthy development of the global pig industry [1].

PRRSV is an enveloped, single-stranded, positive-stranded RNA virus with an un-segmented genome about 15 kb in length and containing 11 open reading frames. The 11 open reading frames are ORF1a, ORF1b, and ORF2-7, and there are also the newly identified ORF (TF) and −1/−2 programmed ribosome shift signals that express two novel proteins, Nsp2TF and Nsp2N. ORF1a and ORF1b encode nonstructural proteins (NSPs) involved in PRRSV replication, and ORF2-7 encodes vesicle proteins involved in the assembly of PRRSV particles. ORF1a and ORF1b account for approximately 75% of the entire genome of PRRSV in size, and ORF1a translates to encode a large nonstructural precursor polyprotein, PP1a, which is subsequently hydrolyzed and cleaved into NSP1α, NSP1β, and NSP2-8. ORF1a and ORF1b, in turn, encode another nonstructural precursor polyprotein through a ribosomal shift mechanism PP1ab, which is enzymatically cleaved into 14 non-structural proteins, namely, NSP1α, NSP1β, NSP7α, NSP7β, NSP2-6, and NSP8-12. ORF2-7 mainly encodes PRRSV structural proteins (GP2, GP2b, GP3-4, GP5, GP5a, M, and N), with GP5, M, and N proteins being the viral ones. GP5, M, and N proteins are the major structural proteins, while GP2, GP2b, GP3-4, and GP5a are the minor structural proteins. PRRSV belongs to the family *Arteriviridae* and the order *Nidovirales*. It is currently classified into two serotypes based on serological differences: the European type (PRRSV-1) and American type (PRRSV-2) [2].

PRRSV has a strict cytotropic nature, mainly invading porcine alveolar macrophages, and mediating entry into cells through receptors [3]. PRRSV infection tends to cause immunosuppression and a strong inflammatory response. Studies have shown that PRRSV infection can cause apoptosis of thymocytes, resulting in thymus atrophy in piglets [4,5]. According to the current classification system, PRRSV has been reclassified as PRRSV-1 and PRRSV-2. PRRSV-1 was first reported in Europe and later spread throughout the continent. It is currently widespread in countries such as China, the United States, Canada, South Korea, and Thailand. PRRSV-2, represented by the North American strain VR-2332, exhibits only 50–60% nucleotide homology with the PRRSV-1 genotype [2]. When PRRSV infects the host, associated non-structural proteins disrupt the innate immune response, resulting in a weakened acquired immune response. It has been found that PRRSV Nsp1, 2, 4, 7, 11, and N proteins can inhibit the production of type I interferons (IFNs), or affect interferon signaling pathways, thereby escaping the host’s immune surveillance. Nsp1α and Nsp1β inhibit the activation of the IFN-I promoter by hindering the protein post-translational modification of interferon regulatory factor 3 (IRF-3) as well as the phosphorylation and nuclear translocation processes, respectively. Nsp2 inhibits IFN-I production and induces inflammation by bidirectional regulation of the NF-κB signaling pathway in the early and late stages of infection, evading host innate immunity. Nsp4 is the main proteolytic enzyme for PRRSV that inhibits the expression of IFN-I promoters by cleaving the linker molecule MAVS. Nsp11 degrades MAVS mRNA, thereby inhibiting its downstream signaling pathway and impeding IFN-β production [6]. PRRSV infection often leads to poor, late, and irregular activation of the innate and adaptive immune response [7]. Inhibition of IFN-β and upregulation of IL-10 result in severe host immunosuppression.

Macrophages are phagocytic antigen-presenting cells that reside in all tissues. As antigen-presenting cells, macrophages exhibit cross-talk between the innate and adaptive immune systems [8]. Macrophages are an innate immune cell type that express diverse antigen-presenting molecules, including MHC class I and II molecules, as well as co-stimulatory B7.1 (CD80), B7.2 (CD86), and CD40 molecules, among others [9,10]. These molecules play vital roles in the immune regulation of pulmonary alveolar macrophages and in the clearance of respiratory pathogens. Moreover, other related proteins such as natural resistance-associated macrophage protein (Nramp) and pulmonary surfactant protein A (SP-A) perform immunological functions [11,12]. Astragaloside-IV (AS-IV), a saponin active compound, is among the key compounds found in astragalus aqueous extract known for its wide range of pharmacological effects. These effects include reducing oxidative stress, providing cardioprotective benefits, treating inflammation, offering antiviral and antibacterial treatments, alleviating fibrosis, diabetes, and also modifying the body’s immune responses [13,14,15,16,17,18].

The innate immune system is the body’s initial defense against pathogenic micro-organisms. It can identify different related molecular patterns (PAMPs) of invading viruses by encoding various pattern recognition receptors (PRRs) [19]. These PRRs include the toll-like receptor (TLR), C-type lectin receptor (CLR), NOD-like receptor (NLR), RIG-I-like receptor (RLR), and cytosolic DNA receptor (CDR), which primarily mediate gene expression. In recent years, CDR have been newly discovered. The cGAS-stimulator of interferon genes (STING) pathway is mainly responsible for sensing and identifying the abnormal presence of double-stranded DNA in the cytoplasm [20,21], and establishing an effective natural immune response by inducing the expression and secretion IFN-I and interferon-stimulating genes, which is one of the basic mechanisms of the host defense of organisms. cGAS is a cytosolic DNA pattern recognition receptor that directly recognizes and binds dsDNA, catalyzing GTP and ATP to form a secondary messenger 2′3′-cyclic-GMP-AMP (cGAMP). cGAMP then binds to the linker receptor protein STING located in the endoplasmic reticulum membrane, initiating STING dimer conformational changes and, thus, activating. With the transport of STING proteins along the endoplasmic reticulum–endoplasmic reticulum Golgi intermediate (ERGIC)–Golgi apparatus, STING gradually polymerizes and aggregates and recruits the key kinase TANK-binding kinase 1 (TBK1), transcription factor IRF-3, and numerous regulatory proteins to form functional signaling complexes and lead to phosphorylation of IRF-3. The activated IRF-3 dimer is incorporated into the nucleus [22], promoting the secretion and expression of type I interferon and interferon-stimulated genes, thereby defending against invasion by pathogenic micro-organisms and maintaining a state of tissue homeostasis. cGAS not only identifies free abnormal DNA in the cytoplasm, but is also involved in the infection process of RNA viruses. Studies have shown that overexpression of cGAS and stable expression of STING can inhibit the replication of PRRSV. At the same time, destabilizing mitochondria after PRRSV infection will also expose mitochondrial DNA (mtDNA) to the cytoplasm; cGAS recognizes mtDNA, promotes IFN-I expression, and inhibits PRRSV replication [20].

PRRS represents one of the immunosuppressive diseases, and scientific investigations suggest that the cGAS-STING pathway plays a vital role in the PRRSV infection process [23]. The objective of this study was to investigate the molecular mechanism by which AS-IV regulates the cGAS-STING signaling pathway and suppresses innate immunity after PRRSV infection. This research aims to enhance the current understanding of immune evasion from PRRS and offer insights into the development of novel antiviral therapies.

## 2. Materials and Methods

### 2.1. Cells and Virus

Porcine alveolar macrophages (PAM) cells were isolated from PRRSV, PCV, and CSFV negative pigs (Bluefbio, Shanghai, China), and cultured in Dulbecco’s modified Eagle’s medium (DMEM) containing 10% fetal bovine serum (FBS; Gibco, Thermo Fisher Scientific, Waltham, MA, USA) at 37 °C and under 5% CO_2_.

The PRRSV strain JL/07/SW (PRRSV-2, gifted by the Laboratory of Animal Infectious Diseases, College of Animal Medicine, Jilin University), viral titer of 10^6.25^ TCID_50_/0.1 mL was determined as TCID_50_ by endpoint dilution assay. Virus titer was calculated using the Reed and Muench method [23].

### 2.2. Cell Culture and Treatment

#### 2.2.1. In Vitro Cytotoxicity Assay

PAM cells were cultured in 96-well plates at a density of 5 × 10^3^ cells/well, and, after incubation for 24 h, the cells were treated with different concentrations (200, 100, 50, 25, 12.5, and 6.25 μg/mL) of AS-IV (Chengdu Must Bio-Technology Co. Ltd., Chengdu, China) cultured at 5% CO_2_, 37 °C for 48 h. Cell viability was assessed by employing a CCK8 kit (Absin, Shanghai, China) per the manufacturer’s guidelines.

#### 2.2.2. In Vitro Infection Inhibition Assay

PAM cells were inhibited by inoculating 1 × 10^6^ into a 6-well plate. and incubated at 37 °C in a 5% CO_2_ incubator until they formed a monolayer. After a period of 2 h of incubation in 100 TCID50 of PRRSV (500 µL/well), the supernatant was removed, and 2 mL of AS-IV solution (at 50, 25, or 12.5 μg/mL) was added to the corresponding wells. Following this, the cells were incubated for 48 h at 37 °C in 5% CO_2_. Thereafter, the supernatant was collected, and cell viability was determined using the CCK8 assay. Subsequently, the viral inhibition rate was calculated, and the PRRSV load in the supernatant was quantified using qRT-PCR.

#### 2.2.3. Preparation of Positive Standard

The N protein of PRRSV JL/07/SW strain was sequenced and the plasmid was synthesized and constructed by Changchun Kumei organism. After the recombinant plasmid was identified as positive by conventional PCR, the concentration of positive plasmid was determined by ultramicro spectrum analyzer (q6000+). According to the following formula, the concentration was converted into copy number, and then diluted by gradient of 10^−1^ to 10^−9^. The diluted plasmid was used as the standard template and stored at −20 °C. The copy numbers of the plasmids were calculated according to the formula: Copy Number (Copies/μL)  =  Concentration (g/μL)/(660  ×  DNA length)  ×  NA (NA: Avogadro constant) [20]. To identify positive and negative control templates for qRT-PCR amplification, a standard template was diluted from 109 pfu/mL to 102 pfu/mL and used as the positive template, while double-distilled water was employed as the negative control. The results of the standard curve were analyzed and drawn by qRT-PCR software.

### 2.3. ELISA

The cell supernatant was collected and the expression of cytokines, PAM innate immune function molecules, and inflammatory cytokines in the supernatant were detected by ELISA. ELISA kits were utilized per manufacturers’ instructions: IFN-β Pig ELISA kit, cGAMP Pig ELISA kit, Swine Leucocyte Antigen -Ι (SLA-Ι) ELISA Kit, Swine Leucocyte Antigen-Ⅱ (SLA-Ⅱ) ELISA Kit, Cluster of differentiation 80 (CD80) Pig ELISA Kit, Cluster of differentiation 86 (CD86) Pig ELISA Kit, Surfactant Proteins A (SP-A) Pig ELISA Kit, and Natural resistance-associated macrophage protein (Nramp) Pig ELISA Kit (LanpaiBIO, Shanghai, China).

### 2.4. qRT-PCR

Total RNA was extracted using the RNeasy Mini Kit (Qiagen, Hilden, Germany) according to the manufacturer’s instructions. Reverse transcription reactions were conducted using the M-MLV reverse transcription-polymerase system (TaKaRa, Dalian, China) at 25 °C for 5 min followed by 42 °C for 1 h. SYBR Premix Ex Taq™ (Takara) was used to measure the expression of cGAS-STING-pathway-related genes and natural immune function genes of PAM cells. The relative expression levels were determined using the 2^−∆∆Ct^ method [24], with GAPDH mRNA as a reference. See Table 1 for primer details.

### 2.5. Western Blot

To prepare cell lysates, cells were incubated in RIPA buffer with 1 mM phenylmethyl-sulfonyl fluoride and 1 mg/mL of protease inhibitor cocktail (Roche) for 15 min at 4 °C. After centrifuging at 10,000 rpm/min for 10 min, the supernatants were collected, mixed with 5 × sodium dodecyl sulfate-polyacrylamide gel electrophoresis (SDS-PAGE) sample loading buffer (Beyotime), and boiled for 5 min. The proteins were separated using SDS-PAGE and transferred onto a nitrocellulose membrane.

Place nitrocellulose membrane into 5% skim milk and block it for 2 h at room temperature. The membrane was then washed three times with Tris-buffered saline containing 0.1% Tween 20 and subsequently incubated at room temperature with horseradish peroxidase-conjugated goat anti-mouse/rabbit IgG (H + L) antibody (diluted at 1:5000) for 1 h. Following visualization with the Pierce ECL WB substrate (Thermo Fisher Scientific), the membranes were analyzed to determine protein quantification. Levels of target proteins were normalized to the levels of GAPDH. The expression of N protein, cGAS, TBK1, IRF-3, STING, and IFN-β proteins was evaluated through western blot analysis. All protein antibodies were purchased from Wuhan Servicebio Technology Co., Wuhan, China.

### 2.6. Statistical Analysis

All experiments were replicated independently three times. Statistical significance was evaluated using *t*-tests. Results are presented as means ± standard deviations (SD) of the three independent experiments. Statistical significance was set at *p* < 0.05.

## 3. Results

### 3.1. Maximum Nontoxic Concentration Determination

To determine the maximum non-toxic concentration (MNTC) of AS-IV, we assessed the survival rate of PAM cells after incubation with varying concentrations of AS-IV. The results, shown in Figure 1, demonstrate that PAM cells tolerated AS-IV well, with an MNTC of 200 µg/mL. Therefore, for subsequent experiments, we selected concentrations of 50 µg/mL, 25 µg/mL, and 12.5 µg/mL.

### 3.2. Virus Inhibition Assay

Virus inhibition was measured by the qRT-PCR. We first plotted a standard curve using the positive standard (r = 0.99). The effect of different doses of AS-IV on virus proliferation was subsequently examined. The results of the virus inhibition assay showed that all doses of AS-IV could inhibit the proliferation of PRRSV (Figure 2A–C). We determined IFN-β secretion by ELISA and showed that PRRSV infection of PAM cells resulted in a significant decrease in IFN-β expression, while AS-IV significantly restored its secretion (Figure 2D). Therefore, we conducted a follow-up experiment.

### 3.3. Effect of AS-IV on Innate Immune Function after PRRSV Infection

In this section, our focus was on investigating the antigen-presenting and innate immune function molecules of PAM cells using the ELISA and qRT-PCR assays. Firstly, we evaluated the antigen presentation function, whereby SLA-I, SLA-DRB1(SLA-Ⅱ), CD86 and CD80 secretion, and mRNA expression were significantly upregulated upon PRRSV infection. On the other hand, AS-IV treatment led to a significant improvement in the antigen presentation function of PAM cells (Figure 3A–D and Figure 4A–D). Subsequently, we assessed the innate immune function of PAM cells, and our results revealed that PRRSV infection led to an increase in the secretion and mRNA expression of Nramp and SP-A. Notably, AS-IV significantly enhanced innate immune function following PRRSV infection (Figure 3E,F and Figure 4E,F).

### 3.4. AS-IV Regulates the cGAS-STING Signaling Pathway after PRRSV Infection

We examined the cGAS-STING signaling pathway of PAM infection with PRRSV by qRT-PCR. The results show that cGAS and TBK1 mRNA expression decreased after PRRSV infection. AS-IV significantly restores the expression of cGAS and TBK1 mRNA (Figure 5A,B). IRF-3 mRNA expression was significantly elevated after PRRSV infection, while AS-IV significantly decreased its expression (Figure 5C). The expression of STING was activated and was more significantly elevated by AS-IV treatment (Figure 5D). IFN-β secretion was previously detected by ELISA and was found to be reduced after PRRSV infection, which was confirmed here by mRNA expression (Figure 5E). Later, the mRNA expression of OAS1 and ISG15 was measured, and the results showed that PRRSV infection resulted in reduced mRNA expression of OAS1 and ISG15, while AS-IV significantly restored their expression (Figure 5F,G). In addition, we also measured the expression of cGAMP by ELISA. The results showed that the secretion of cGAMP was significantly decreased after PRRSV infection; AS-IV can significantly restore its secretion (Figure 5H).

In here, we detected the expression of some proteins in the cGAS-STING pathway by western blot. The results showed the protein expression of cGAS, TBK1, and IFN-β was significantly decreased after PRRSV infection; however, AS-IV could restore their expression (Figure 6A–C,E). IRF-3 protein expression was significantly increased after PRRSV infection, and AS-IV could inhibit its expression (Figure 6D). The activation of STING protein occurs after PRRSV infection, and its expression can be increased significantly with AS-IV treatment (Figure 6F).

## 4. Discussion

The PRRS has a substantial economic impact on the global swine industry, making it a highly significant issue [25]. Currently, no specific treatment exists for PRRS, and thus vaccination remains an effective strategy for disease control [26]. However, due to the immune evasion properties of PRRSV, the PRRSV vaccine offers limited protection against the disease. Therefore, finding new strategies to mitigate the economic impact of PRRS can greatly benefit the pig industry.

IFN-I has a very important role in the body’s antiviral immunity and serves as a bridge between innate and adaptive immunity [27,28]. In PRRSV infections, with macrophages as PRRSV-trophobic cells, it is the macrophages that play a critical factor in the ability of the pathogen to subvert host immunity and proliferate PRRSV [29]. Infected macrophages recognize viral RNA through pattern recognition receptors on their surface or cytoplasm, which, in turn, activate a series of downstream antiviral signaling and induce the production of large amounts of IFN-I. IFN-I induces the synthesis of antiviral proteins, on the one hand, and, on the other hand, IFN-I further activates uninfected cells to secrete large amounts of ISGs by cellular autocrine or paracrine means, allowing uninfected cells to establish antiviral status in advance. IFN-β is an important molecule in the innate immunity against viruses [30]. In this study, we employed primary alveolar PAM cells for several experiments. Multiple studies have suggested that most DNA viral proteins directly modulate cGAS/STING signaling by targeting pivotal proteins including cGAS, STING, and TBK1 [31].

PRRSV, as an RNA virus, has also been reported to have the cGAS-STING signaling pathway involved in its infection process [32]. PRRSV is known to be highly immunosuppressive, one manifestation of which is a significant decrease in IFN-β expression following PRRSV infection. However, the cGAS-STING pathway is crucial for IFN-β production by activating IRF-3 [22]. AS-IV is purified from the Chinese medicinal herb; studies have shown that AS-IV has biological activities, including anti-oxidant, anti-inflammatory, anti-virus, anti-aging, and anti-platelet aggregation activities [16]. Taking these backgrounds into consideration, we tested the inhibitory effects of AS-IV on PRRSV infection and explores its proof-of-principle mode of action in vitro. Our data show that treatment with AS-IV reduces viral load and titer (Figure 2A–C), restores IFN-β expression (Figure 2D, Figure 5E and Figure 6E), possibly by restoring repressed cGAS and TBK1 expression (Figure 5A,B and Figure 6B,C), restores interferon-stimulated genes OAS1 and ISG15 expression (Figure 5F,G), downregulates elevated IRF-3 (Figure 5C and Figure 6D), activates STING expression (Figure 5D and Figure 6F) increases cGAMP secretion (Figure 5H), and, finally, restores the cGAS-STING signaling pathway. The signaling activity of proteins such as IRF-3 and TBK1 is primarily regulated by phosphorylation events rather than the absolute protein amounts. Phosphorylation plays a crucial role in modulating the activation and downstream signaling of these proteins. In the immune response, the phosphorylation of TBK1 is mediated by other activated protein kinases. For example, during viral infection, TBK1 can interact with virus-associated pattern recognition receptors (such as RIG-I and TLR3) and become activated. Phosphorylated TBK1 exhibits enhanced kinase activity and can regulate multiple downstream signaling pathways. One of the primary functions of TBK1 is the phosphorylation of IRF3. Phosphorylated IRF3 can enter the cell nucleus and bind to DNA, activating the transcription of interferons and other immune-related genes, thus initiating antiviral and immune responses. Upon phosphorylation and activation, IRF-3 undergoes conformational changes and forms dimers or multimers. These activated IRF-3 complexes can enter the cell nucleus and bind to DNA, initiating the transcription of interferon-beta and other immune-related genes. The phosphorylation modifications of TBK1 and IRF-3 play important roles in the cellular response to infection, inflammation, and other stressful stimuli. In our study, we focused on assessing the protein amounts of these signaling molecules as a first step in understanding their expression patterns and potential modulation under the conditions being investigated. However, the protein phosphorylation status is an important aspect of their functional regulation and should be considered.

In this study, we additionally measured the effect of AS-IV on the innate immune function of PAM cells after PRRSV infection The PRRS virus (PRRSV) targets antigen-presenting cells, such as macrophages and dendritic cells, which are crucial components of the innate immune system. Accordingly, it has complex effects on both the innate and adaptive immune responses. PRRSV evokes a lengthy viremia by impeding the immune response in its entirety, primarily characterized by heightened pro-inflammatory and immunomodulatory responses, low antiviral responses, and impaired cellular responses [32,33]. Several studies have found that PRRSV infection leads to the reduced mRNA expression of antigen-presenting functions in PAM cells by measuring the mRNA transcriptome [34], which was also well-verified in our experiments (Figure 4A–F).

The immune system response is governed by various molecules necessary to defend the body against foreign bodies and danger signals while still maintaining tolerance to self-antigens [35]. Macrophages are considered to be pleiotropic phagocytes with a crucial role in both innate and adaptive immunity. The primary duties of macrophages include chemotaxis, phagocytosis, endocytosis, antigen presentation, and cytokine secretion, profoundly impacting immune responses [36].

The swine major histocompatibility complex (MHC) is a vast multigene family responsible for encoding swine leukocyte antigens (SLA) that are glycoproteins found on the cell surface membrane. These SLA molecules are classified into three regions: SLA class I, II, and III. The main function of SLA-I is to present antigen peptides to CD8+T cells, activate them, and release cytotoxins to kill infected cells. The main function of SLA-II is to present exogenous antigens to CD4+T cells, and activate them to participate in immune responses [37,38]. Therefore, in this study, we used ELISA and qRT-PCR to detect the secretion and mRNA expression levels of SLA molecules in PAM cells after PRRSV infection. The results showed that the expression of SLA molecules was significantly inhibited after PRRSV infection, while AS-IV can significantly restore its expression (Figure 3A,B and Figure 4A,B).

Macrophages express pro-inflammatory or anti-inflammatory co-stimulatory molecules on their surface which belong to the B7 family, comprising 10 reported members such as CD80 (B7-1), CD86 (B7-2), B7-DC, PD-L1 (B7-H1), B7-H2, B7-H3, B7-H4, B7-H5, B7-H6, and B7-H7 [39,40]. The B7 receptor present on the T-cells surface is CD28/cytotoxic T lymphocyte antigen-4 (CTLA-4). The interaction between B7 and CD28 promotes T-cell activation and proliferation, and also regulates T-helper (TH)1/TH2 cell differentiation, and contributes to B-cell antibody production and isotype switching, in addition to promoting cytokine secretion [41]. In contrast, the interaction between B7 and CTLA-4 inhibits T-cell activation and proliferation [42,43]. In this section of the study, we employed the ELISA and qPCR techniques to examine the secretion and mRNA expression of CD80 and CD86 in PAM cells following PRRSV infection. Our findings demonstrated that the expression of CD80 and CD86 was considerably suppressed after PRRSV infection, whereas AS-IV treatment significantly restored their expression (refer to Figure 3C,D and Figure 4C,D). Alveolar macrophages also express other molecules related to the innate immune function of the lung, such as Nramp1 and SP-A. These are important non-specific immune defense molecules in the lung that not only clear pathogens but also participate in the regulation of lung immune, inflammatory, and allergic responses. NRAMP plays a crucial role in the immune response against intracellular pathogens [44]. Specifically, the NRAMP1 protein targets the membrane of microbe-containing phagosomes and can modify the intra-phagosomal environment to influence microbial replication [45]. This family of genes may enable the organism to develop resistance to disease infestation through the transport of metal ions. SP-A is involved in lung innate host defense and surfactant-related functions [46,47]. SP-A downregulates the human alveolar macrophage pro-inflammatory cytokines such as tumor necrosis factor-alpha (TNF-α), interleukin 1-beta (IL-1β), and macrophage inflammatory protein 1 alpha (MIP-1α) in a dose-dependent manner. In this section of the study, we investigated the secretion and mRNA expression of Nramp and SP-A in PAM cells after PRRSV infection. Our results exhibited notable downregulation in the expression of Nramp and SP-A post-PRRSV infection, whereas treatment with AS-IV significantly restored their expression (Figure 3E,F and Figure 4E,F). PRRSV infection leads to reduced natural immune function in PAM cells, including diminished antigen-presentation function and reduced expression of innate immune molecules (Figure 3 and Figure 4).

In contrast, AS-IV can suppress this response well, suggesting, on the other hand, that AS-IV can alleviate the immunosuppression caused by PRRSV infection. Overall, this study shows that AS-IV can attenuate well the immunosuppression that occurs with PRRSV infection, by detecting the expression of the cGAS-STING signaling pathway and the natural immune function of PAM cells. This result suggests potential new antiviral strategies.

In vitro cell culture experiments are typically conducted in controlled artificial environments with limited external stimuli. In contrast, the immune response in the entire organism is regulated by multiple complex factors, including cell–cell interactions, cell–matrix interactions, and hormone regulation. Furthermore, there are interactions and regulations among multiple tissues and organs in the entire organism. These cell–cell and tissue–tissue interactions can influence signal transduction, metabolic pathways, and drug absorption, distribution, and metabolism. Therefore, there may be certain differences between in vitro results and in vivo effects in the whole organism. Although in vitro results cannot fully simulate the biological environment of the entire organism, in vitro experiments are still important tools for studying disease mechanisms and evaluating the efficacy of drug candidates. In vitro experiments can provide initial insights into specific cell types and molecular mechanisms and offer clues for further in vivo studies. To better explain the potential differences between in vitro results and in vivo effects, our future experiments may include the use of animal models and clinical trials to comprehensively assess the relevance and feasibility of research findings in the entire organism.

## 5. Conclusions

AS-IV can effectively reduce PRRSV replication, restore the cGAS-STING pathway and innate immune function that were suppressed after PRRSV infection, and exert antiviral effects.

## Figures and Tables

**Figure 1 viruses-15-01586-f001:**
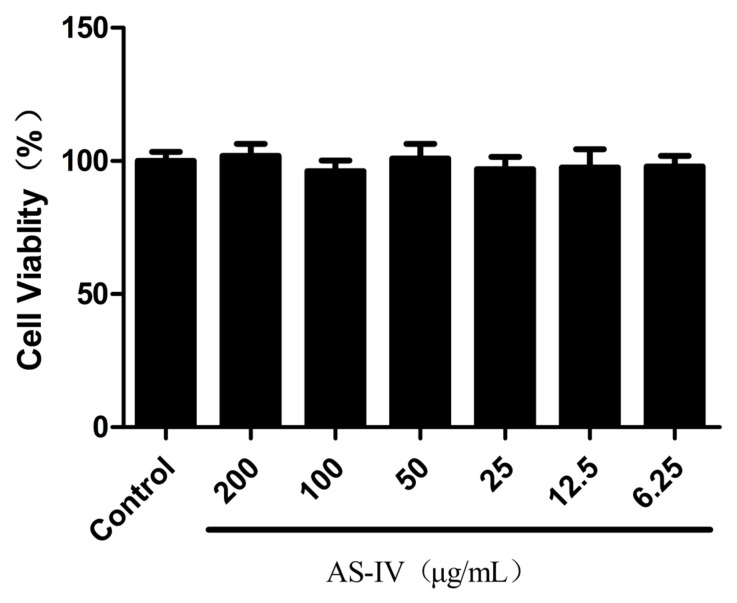
The cytotoxic effect of AS-IV on PAM cells was detected by CCK8 method.

**Figure 2 viruses-15-01586-f002:**
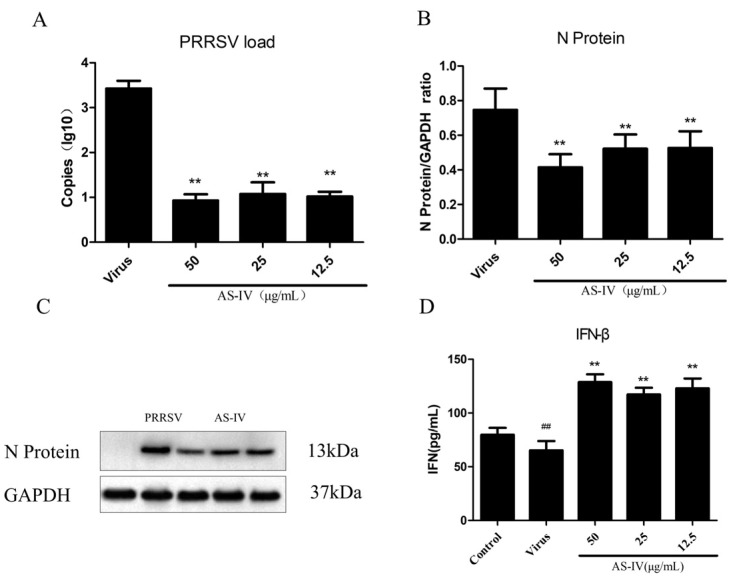
Effect of AS-IV antiviral: (**A**) PRRSV load; (**B**) N protein expression; (**C**) Western blot; (**D**) IFN-β secretion. Data are presented as mean ± SD. ^##^ *p* < 0.01, vs. the control. ** *p* < 0.01, vs. the virus group.

**Figure 3 viruses-15-01586-f003:**
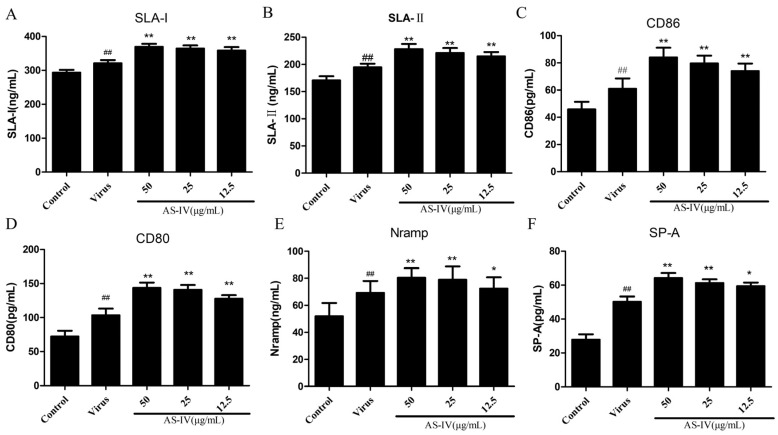
Innate immune function of PAM cells after infection with PRRSV: (**A**) SLA-I secretion; (**B**) SLA-Ⅱ secretion; (**C**) CD86 secretion. (**D**) CD80 secretion. (**E**) Nramp secretion. (**F**) SP-A secretion. Data are presented as mean ± SD. ^##^ *p* < 0.01, vs. the control. ** *p* < 0.01, * *p* < 0.01, vs. the virus group.

**Figure 4 viruses-15-01586-f004:**
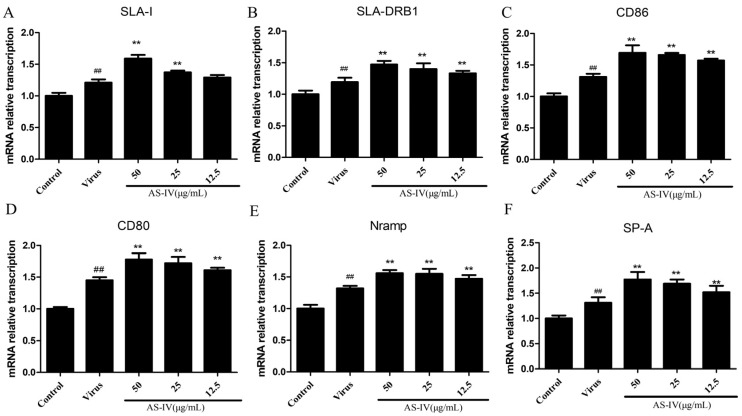
Relative changes in mRNA expression of innate immune function: (**A**) SLA-I mRNA expression. (**B**) SLA-DRB1 mRNA expression. (**C**) CD86 mRNA expression. (**D**) CD80 mRNA expression. (**E**) Nramp mRNA expression. (**F**) SP-A mRNA expression. Data are presented as mean ± SD. ^##^ *p* < 0.01, vs. the control. ** *p* < 0.01, vs. the virus group.

**Figure 5 viruses-15-01586-f005:**
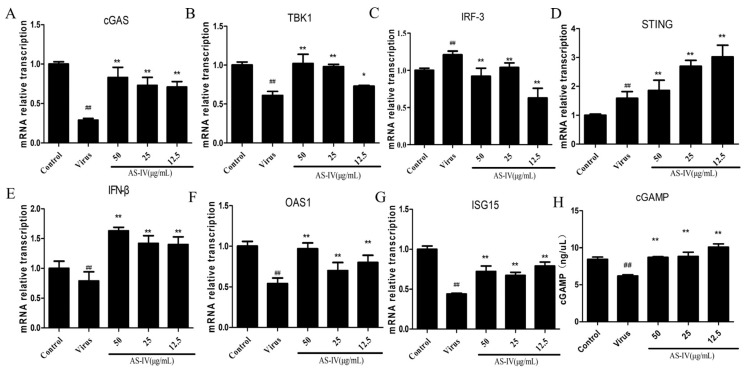
Relative changes in mRNA expression of c GAS-STING signal pathway: (**A**) cGAS mRNA expression. (**B**) TBK1 mRNA expression. (**C**) IRF-3 mRNA expression. (**D**) STING mRNA expression. (**E**) IFN-β mRNA expression. (**F**) OAS1 mRNA expression. (**G**) ISG15 mRNA expression. (**H**) cGAMP secretion. Data are presented as mean ± SD. ^##^ *p* < 0.01, vs. the control. ** *p* < 0.01, * *p* < 0.01, vs. the virus group.

**Figure 6 viruses-15-01586-f006:**
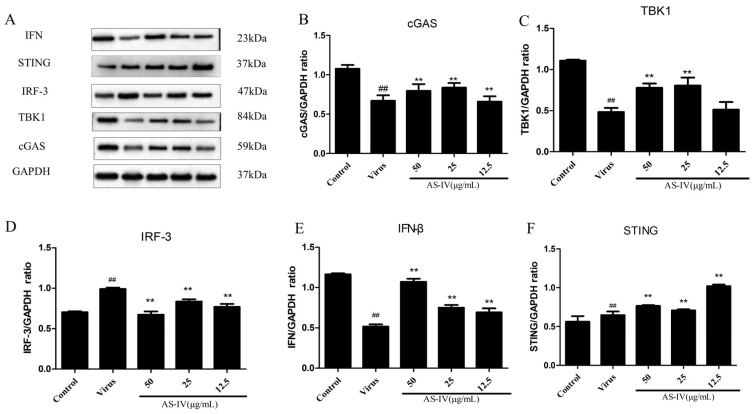
AS-IV regulates relative changes in protein expression of cGAS-STING signal pathway: (**A**) western blotting analysis of protein expression. (**B**) cGAS protein expression. (**C**) TBK1 protein expression. (**D**) IRF-3 protein expression. (**E**) IFN-β protein expression. (**F**) STING protein expression. Data are presented as mean ± SD. ^##^ *p* < 0.01, vs. the control. ** *p* < 0.01, vs. the virus group.

**Table 1 viruses-15-01586-t001:** Sequences of primers.

Name	Sequence (5′—3′)
GAPDH-F	TGACATCAAGAAGGTGGTGAAGCAG
GAPDH-R	GTGTCGCTGTTGAAGTCAGAGGAG
cGAS-F	CTTTCACCTATGTGCCGACAACCC
cGAS-R	TGTCACGCAGTTATCAAAGCAGAGG
TBK1-F	AAGCCTTCTGGTGCAATATCTGGAG
TBK1-R	ACCTGAAGACCCCGAGAAAGACTG
IRF-3-F	GAGGCTCGTGATGGTCAAGGTTG
IRF-3-R	AGTGGGTGGCTGTTGGAAATGTG
IFN-F	GAGTGTGGAGACCATCAAGGAAGAC
IFN-R	GTTCATGTACTGCTTTGCGTTGGAC
OAS1-F	GCGAGTTCTCCACCTGCTTCAC
OAS1-R	ACTAGGCGGATGAGGCTCTTGAG
ISG15-F	GGTGGTGGACAGATGCGATGAAC
ISG15-R	GGCTCACTTGCTGCTTCAGGTG
SLA-1-F	GGATGAGGAGACGCGGAAAGTCA
SLA-1-R	TGGTCCCAAGTAGCAGCCAAACA
SLA-DRB1-F	CGACTTTGACCCGCAGAATGG
SLA-DRB1-R	TGGTGGCTCTGGTATGGTTGGA
CD86-F	CTCTTTGTGATGGTCCTCCTG
CD86-R	AGGCTTAGGTTCTGCGAGTT
CD80-F	GGGAACACCATTACCCAAGC
CD80-R	GTCACCTGAACGATGCCTGA
Nramp1-F	GTCTCCTTCTTCATCAACCTCTT
Nramp1-R	ATCACGCCGCCTTGGTAA
SP-A-F	GGAGACTTCTTCTACTTGGATGG
SP-A-R	GCTGGCAGTTCCTGTCATTC

## Data Availability

The datasets generated for this study are available from the corresponding author upon request.

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
