# Peer review of "Astragaloside IV Regulates cGAS-STING Signaling Pathway to Alleviate Immunosuppression Caused by PRRSV Infection"

_viruses, 2023, doi:10.3390/v15071586_

Round 1

Reviewer 1 Report

This manuscript examines the effects of AS-IV treatment of PAMs infected with a strain of PRRSV-2. Specifically, aspects of immune regulation responses are investigated.

Specific comments:

1- In the introduction, the two species of PRRSV must be mentioned.

2- In section 2.1, it must be specified whether the virus is PRRSV-1 or PRRSV-2.

3- In section 2.2.2, please explain how primary PAMs are able to form a monolayer since they do not replicate.

4- Some references throughout the text are not displayed correctly.

5- In the discussion, it would be useful to address potential differences between in vitro results based on cell culture models with possible in vivo effects in the whole organism.

Some editing of the English language is necessary to improve readability of the manuscript.

Author Response

Response to reviewer comments

Thanks to the reviewers for their valuable comments, which we have responded to here, and revised in response to the comments.

Q1: In the introduction, the two species of PRRSV must be mentioned.

A:Revisions have been made in the manuscript.

Q2:In section 2.1, it must be specified whether the virus is PRRSV-1 or PRRSV-2.

A:Revisions have been made in the manuscript.

Q3:In section 2.2.2, please explain how primary PAMs are able to form a monolayer since they do not replicate.

A:The PAMs used in this study were purchased cell lines that could be cultured for passaging.

Q4:Some references throughout the text are not displayed correctly.

A: Revisions have been made in the manuscript.

Q5:In the discussion, it would be useful to address potential differences between in vitro results based on cell culture models with possible in vivo effects in the whole organism.

A: Thank you for your valuable feedback, We agree that discussing this aspect would provide a more comprehensive understanding of the research findings and their implications.

In the revised discussion section, we will incorporate a discussion on the limitations of in vitro models and the importance of considering in vivo effects. We will highlight the need for further studies using animal models or clinical trials to validate and translate the findings observed in cell culture models to the complex physiological conditions of the whole organism. This will help to provide a more accurate assessment of the potential implications and applicability of our research in real-life scenarios.

Reviewer 2 Report

A manuscript (viruses-2474611) entitled “Astragaloside IV regulates cGAS-STING signaling pathway to alleviate immunosuppression caused by PRRSV infection

 by Ke Song et al is describing the effect of plant product of the genus Astragalus, Astragaloside IV, on the induction of innate immunity pathway, i.e. cyclic GMP-AMP synthase-stimulator of interferon genes (cGAS-STING) signaling pathway, in the pig alveolar macrophages during the PRRSV infection. Expression of the type-1 IFN gene and IFN-related several genes are evaluated by real-time PCR. Protein expression levels were measured by the quantitative ELISA or by the specific band on the western blot. This study shows that PRRSV hampers the expression of IFN beta and interferon-related proteins. In addition, Astragaloside IV is shown to enhance the expression of IFN beta and interferon-related protein in the PRSSV-infected pig alveolar macrophage and inhibited the replication of PRRSV. This finding may provide new anti-PRRSV drug candidates, and worth to be considered for publication in Viruses. However, this reviewer would like to rise one major comment and several minor comments and suggestions before accepting this manuscript. 

Major comment

This reviewer has serious concerns about the porcine alveola macrophage (PAM) cells that were used in this study. Because non-infected and non-treated PAM cells are already expressing several amounts of type-I IFN (Figure 2D). It is said that interferons alpha/beta are stimulated in response to the challenge of host defenses. This means that the amount of type-I IFN of normal (non-stimulated) cells is none or low. This reviewer doubts if commercially available PAM cells are not normal cells but are pre-exposed with some pathogen and if these cells used in this study are not suitable for analyzing the cGAS-STING pathway. This reviewer would like to suggest adding the Poly I:C inoculating group as a positive control to see the full activation of type-I IFN in PAM cells. The combination study of Poly I:C and Astragaloside IV treatment may contribute to understanding the mechanism of how Astragaloside IV regulates the cGAS-STING signaling pathway directly or indirectly by hampering the replication of PRRSV.

Minor comments

1.      Introduction: The introduction section is not a simple explanation to explain the background of this study. This section had better shorten and it should be clarified to focus on the field of this study.

2.      Lines 30-31; An explanation about PRRSV is described as “PRRSV is an enveloped, single-stranded positive-stranded RNA virus with an unsegmented genome about 15 kb in length and containing 11 open reading frames.”, but should show which gene or ORF codes Nsp1, 2, 4, 7, 11 and N protein (line 43). Please clarify the gene structure coded on the PRRSV genome. 

3.      Lines 44, 45, 62, and 86: “IFN-1”, “IRF-3”, “NRAMP” and ”STING” should be spelled out when they are used at first time. As shown in “Instructions for Authors”, Acronyms /Abbreviations should be defined the first time they appear in each of three sections: the abstract; the main text; the first figure or table. When defined for the first time, the acronym /abbreviation should be added in parentheses after the written-out form. On the other hand, see lines 88, 97, 63, and 93. “IFN-1”, “IRF-3”, “NRAMP” and ”STING” are shown with their spell out form. 

4.      Line 57; Please clarify the “innate immune cells” described here. 

5.      Lines 58, 74, 85, and 117; Please delete unnecessary hyphens. “di-verse”, im-munity”, “dis-covered” and “bo-vine”.

6.      Lines 71-73; A sentence describing “qi” and “yang” is unclear. Is this sentence necessary? 

7.      Lines 83-84: Terms “TLR”, “NLR” and “RLR” are described without their spell-out form. Follow the guideline shown in “Instructions for Authors”. They should show a Toll-like receptor (TLR), NOD-like receptor (NLR), and RIG-I-like receptor (RLR). 

8.      Line 100-101: A meaning of the sentence is unclear. Please clarify the following sentence “which promotes type I interferon and interferon to stimulate the expression and secretion of genes,”. 

9.      Line 120: Please delete the minus “-“ from “10 exp-6.25”. 

10.   Line 120: Please explain which type and which lineage PRSRSV strain JL/07/SW belongs to. 

11.   Line 126, 168, 201, 207, 219 and 220, 225, 284, 304: “PAMs” and “PAM cells” (See lines 116, 132, and 196, 197, 215) are used at the same time throughout the text. Please explain the difference. If there is no discrimination, they should unify. 

12.   Line 128: “AS-IV (CHENGDU MUST BIO CO., LTD)” should be shown with city and country like “AS-IV (Chengdu Must Bio-Technology Co. Ltd, Chengdu China)”. 

13.   Line 132: Please show how many PAM cells were used for the infection inhibition assay. 

14.   Line 133: Is the infection dose of 100 TCID50 of PRRSV correct? This reviewer thinks that 100 TCID50 for the cells in 6 well plates is too low to see the cytopathogenic effects of PRRSV because a multiplicity of infection (moi) is estimated to be less than 0.01. 

15.   Lines 133 and 136: A time unit “hours” should be shown as “h” like other sentences. See lines 127 and 129. 

16.   Line 149: “Avogadros constant” had better be shown as “Avogadro constant” or “Avogadro’s constant”. 

17.   Line 150: “from 102 to 109” should be shown correctly. 

18.   Lines 151-152: A sentence starting form “Results” is unclear. Please clarify the sentence. 

19.   Lines 154-161; ELISA: Please show how to quantitate the amount of interferon-beta (Figure 2D). 

20.   Lines 163-170, 176, and 180: The time unit “minute(s)” and “hour(s)” should be shown as “min” and “h”, respectively. 

21.   Table 1: Please explain “gRNA” in the table title. Is gRNA used in the present study?

22.   Table 1: The housekeeping gene primer pair for Gapdh-F and Gapdh-R had better be shown as GAPDH-F and GAPDH-R. 

23.   Line 174: “RIPA” had better be shown as “RIPA buffer”, because RIPA means Radio-Immunoprecipitation Assay and not the solution name. 

24.   Lines 180-188: Please explain the primary antibodies used to detect the GAPDH, PRRSV-N, cGAS, TBK1, IRF-3, STING, and IFN-β proteins. 

25.   Figure 2A and 2B: Copy numbers of PRRSV in AS-IV treated PAM cells are almost 1/100 compared to that of PRRSV in non-treated cells.  On the other hand, the amounts of PRRSV N protein in AS-IV treated PAM cells have no 100 times difference compared to that of PRRSV in non-treated cells, but have only 2 times difference. Please explain why genome copy number of PRRSV does not reflect the amount of PRRSV N protein. 

26.   Figure 2C: Please show the applied sample on the top of lanes to help the better understanding. 

27.   Figures 2D: Please add the explanation of “##” in the caption like “## p < 0.01, vs the control”. 

28.   Figure 6A: Please show the applied sample on the top of lanes to help the better understandings.

29.   Discussion: In this study, the protein amount of several signaling proteins are examined. As far as this reviewer knows, signaling of IRF-3 and TBK1 is not controlled by its amount but by the protein phosphorylation. This study does not examine the protein phosphorylation, but needs at least the discussion. 

30.   Citations in the text should be shown in the normal font. Please refer to the style in other articles on MDPI Viruses.

Moderate editing of English language required. Some of comments are included in the comments and suggestios for authors.

Author Response

Response to reviewer comments

Thank you for your valuable feedback,we have revised the manuscript in response to your comments.

Major comment

Q:This reviewer has serious concerns about the porcine alveola macrophage (PAM) cells that were used in this study. Because non-infected and non-treated PAM cells are already expressing several amounts of type-I IFN (Figure 2D). It is said that interferons alpha/beta are stimulated in response to the challenge of host defenses. This means that the amount of type-I IFN of normal (non-stimulated) cells is none or low. This reviewer doubts if commercially available PAM cells are not normal cells but are pre-exposed with some pathogen and if these cells used in this study are not suitable for analyzing the cGAS-STING pathway. This reviewer would like to suggest adding the Poly I:C inoculating group as a positive control to see the full activation of type-I IFN in PAM cells. The combination study of Poly I:C and Astragaloside IV treatment may contribute to understanding the mechanism of how Astragaloside IV regulates the cGAS-STING signaling pathway directly or indirectly by hampering the replication of PRRSV.

A:In this section we added the Poly I:C inoculation group as a positive control to observe the complete activation of type I IFN in PAM cells. The results are as follows.

Our results show that IFN secretion is significantly elevated by Poly I:C treatment, whereas PRRSV infection inhibits IFN secretion. In contrast, AS-IV co-treatment with Poly I:C significantly increased IFN secretion. It was demonstrated that AS-IV could increase the level of IFN secretion in PAMs.

Minor comments

Q1:Introduction: The introduction section is not a simple explanation to explain the background of this study. This section had better shorten and it should be clarified to focus on the field of this study.

A:Revisions have been made in the manuscript.

Q2:Lines 30-31; An explanation about PRRSV is described as “PRRSV is an enveloped, single-stranded positive-stranded RNA virus with an unsegmented genome about 15 kb in length and containing 11 open reading frames.”, but should show which gene or ORF codes Nsp1, 2, 4, 7, 11 and N protein (line 43). Please clarify the gene structure coded on the PRRSV genome.

A: Revisions have been made in the manuscript.

Q3.Lines 44, 45, 62, and 86: “IFN-1”, “IRF-3”, “NRAMP” and ”STING” should be spelled out when they are used at first time. As shown in “Instructions for Authors”, Acronyms /Abbreviations should be defined the first time they appear in each of three sections: the abstract; the main text; the first figure or table. When defined for the first time, the acronym /abbreviation should be added in parentheses after the written-out form. On the other hand, see lines 88, 97, 63, and 93. “IFN-1”, “IRF-3”, “NRAMP” and ”STING” are shown with their spell out form.

A: Revisions have been made in the manuscript.

Q4:Line 57; Please clarify the “innate immune cells” described here.

A:innate immune cells is a class of cells belonging to the innate immune system that is primarily responsible for the initial defense against invading pathogens.

Q5. Lines 58, 74, 85, and 117; Please delete unnecessary hyphens. “di-verse”, im-munity”, “dis-covered” and “bo-vine”.

A:Revisions have been made in the manuscript.

Q6:Lines 71-73; A sentence describing “qi” and “yang” is unclear. Is this sentence necessary?

A:Revisions have been made in the manuscript.

Q7:Lines 83-84: Terms “TLR”, “NLR” and “RLR” are described without their spell-out form. Follow the guideline shown in “Instructions for Authors”. They should show a Toll-like receptor (TLR), NOD-like receptor (NLR), and RIG-I-like receptor (RLR).

A:Revisions have been made in the manuscript.

Q8:Line 100-101: A meaning of the sentence is unclear. Please clarify the following sentence “which promotes type I interferon and interferon to stimulate the expression and secretion of genes,”.

A:Revisions have been made in the manuscript.

Q9:Line 120: Please delete the minus “-“ from “10 exp-6.25”.

A:Revisions have been made in the manuscript.

Q10:Line 120: Please explain which type and which lineage PRSRSV strain JL/07/SW belongs to.

A:PRSRSV strain JL/07/SW belongs to PRRSV 2,is a PRRSV strain isolated in Jilin Province, China.

Q11:Line 126, 168, 201, 207, 219 and 220, 225, 284, 304: “PAMs” and “PAM cells” (See lines 116, 132, and 196, 197, 215) are used at the same time throughout the text. Please explain the difference. If there is no discrimination, they should unify.

A:Revisions have been made in the manuscript.

Q12: Line 128: “AS-IV (CHENGDU MUST BIO CO., LTD)” should be shown with city and country like “AS-IV (Chengdu Must Bio-Technology Co. Ltd, Chengdu China)”.

A: Revisions have been made in the manuscript.

Q13: Line 132: Please show how many PAM cells were used for the infection inhibition assay.

A: Revisions have been made in the manuscript.

Q14: Line 133: Is the infection dose of 100 TCID50 of PRRSV correct? This reviewer thinks that 100 TCID50 for the cells in 6 well plates is too low to see the cytopathogenic effects of PRRSV because a multiplicity of infection (moi) is estimated to be less than 0.01.

A:In our experimental design, we chose a specific infection dose based on previous literature and preliminary studies. PAMs cells were found to be very sensitive to PRRSV, and 1MOI resulted in a large number of deaths of PAMs, so here a 100 TCID50 infection dose was chosen for the experiment.

Q15:Lines 133 and 136: A time unit “hours” should be shown as “h” like other sentences. See lines 127 and 129.

A:Revisions have been made in the manuscript.

Q16:Line 149: “Avogadros constant” had better be shown as “Avogadro constant” or “Avogadro’s constant”.

A: Revisions have been made in the manuscript.

Q17:Line 150: “from 102 to 109” should be shown correctly.

A: Revisions have been made in the manuscript.

Q18:Lines 151-152: A sentence starting form “Results” is unclear. Please clarify the sentence.

A: Revisions have been made in the manuscript.

Q19:Lines 154-161; ELISA: Please show how to quantitate the amount of interferon-beta (Figure 2D).

A: Revisions have been made in the manuscript.

Q20:Lines 163-170, 176, and 180: The time unit “minute(s)” and “hour(s)” should be shown as “min” and “h”, respectively.

A: Revisions have been made in the manuscript.

Q21: Table 1: Please explain “gRNA” in the table title. Is gRNA used in the present study?

A: We are very sorry, this is an oversight in our work, gRNA was not used in this study and has been revised in the manuscript.

Q22:Table 1: The housekeeping gene primer pair for Gapdh-F and Gapdh-R had better be shown as GAPDH-F and GAPDH-R.

A: Revisions have been made in the manuscript.

Q23:Line 174: “RIPA” had better be shown as “RIPA buffer”, because RIPA means Radio-Immunoprecipitation Assay and not the solution name.

A: Revisions have been made in the manuscript.

Q24:Lines 180-188: Please explain the primary antibodies used to detect the GAPDH, PRRSV-N, cGAS, TBK1, IRF-3, STING, and IFN-β proteins.

A: Revisions have been made in the manuscript.

Q25 Figure 2A and 2B: Copy numbers of PRRSV in AS-IV treated PAM cells are almost 1/100 compared to that of PRRSV in non-treated cells.  On the other hand, the amounts of PRRSV N protein in AS-IV treated PAM cells have no 100 times difference compared to that of PRRSV in non-treated cells, but have only 2 times difference. Please explain why genome copy number of PRRSV does not reflect the amount of PRRSV N protein.

A:Thank you for your question. The difference in PRRSV genome copy numbers and the amounts of PRRSV N protein can be attributed to several factors. Firstly, the viral genome copy number reflects the replication and presence of the viral genetic material within the infected cells. It provides information about the viral RNA replication and the overall viral load. On the other hand, the amount of PRRSV N protein represents the protein expression level, which is influenced by various factors such as viral replication efficiency, transcriptional and translational regulation, protein stability, and post-translational modifications.

Several reasons can contribute to the discrepancy between genome copy numbers and protein expression levels. Firstly, the translation efficiency of viral RNA into proteins can vary due to different factors such as RNA secondary structures, codon usage bias, and the presence of regulatory elements within the viral genome. Additionally, post-translational modifications and protein degradation processes can also affect the stability and accumulation of viral proteins.

It is also important to consider the limitations of the techniques used to measure genome copy numbers and protein expression levels. Quantification of viral genome copies is typically achieved through molecular techniques such as quantitative PCR, while protein expression levels are assessed using methods such as Western blotting . These techniques have their own inherent biases and variations that can contribute to the observed differences.

Q26:Figure 2C: Please show the applied sample on the top of lanes to help the better understanding.

A: Revisions have been made in the manuscript.

Q27:Figures 2D: Please add the explanation of “##” in the caption like “## p < 0.01, vs the control”.

A: Revisions have been made in the manuscript.

Q28:Figure 6A: Please show the applied sample on the top of lanes to help the better understandings.

A: Revisions have been made in the manuscript.

Q29:Discussion: In this study, the protein amount of several signaling proteins are examined. As far as this reviewer knows, signaling of IRF-3 and TBK1 is not controlled by its amount but by the protein phosphorylation. This study does not examine the protein phosphorylation, but needs at least the discussion.

A: Revisions have been made in the manuscript.

Q30:Citations in the text should be shown in the normal font. Please refer to the style in other articles on MDPI Viruses.

A: Revisions have been made in the manuscript.

Round 2

Reviewer 2 Report

The manuscript is properly revised according to the suggestions and comments by the reviewers.

I recommend this manuscript as a suitable paper for accepting the publication in Viruses.

Author Response

Thank you for your approval